# Predictive role of the peripheral blood inflammation indices neutrophil-to-lymphocyte ratio (NLR), platelet-to-lymphocyte ratio (PLR), and systemic immunoinflammatory index (SII) for age-related cataract risk

**Baohua Li**[1☯], **Xinyue Hou**[1☯], **Bobiao Ning**[2☯], **Xiao Li**[3], **MingMing Zhang**[1], **Jianquan Wang**[1], **Mengyu Liu**[1], **Yipeng Shi**[1], **Zefeng Kang**[1]*

**1** China Academy of Chinese Medical Sciences, Eye Hospital, Beijing, China, **2** Department of Dermatology, China Academy of Chinese Medical Sciences Guang'anmen Hospital, Beijing, China, **3** First Clinical Medical College, Shandong University of Traditional Chinese Medicine, Jinan, Shandong, China

☯ These authors contributed equally to this work.

\* Kangzf7717@126.com

**Data Availability Statement:** https://www.cdc.gov/nchs/nhanes/index.htm.

## Abstract

The novel inflammatory markers neutrophil-to-lymphocyte ratio (NLR), platelet-to-lymphocyte ratio (PLR), and systemic immunoinflammatory index (SII) have not yet been used in the study of age-related cataracts. The aim of this study was to investigate the possible relationships between the NLR, PLR, and SII and age-related cataracts. In the 2005–2008 National Health and Nutrition Examination Survey (NHANES) cross-sectional surveys, we collected complete information on blood counts, whether cataract surgery had been performed, and baseline information for adults. We investigated the independent interactions between the inflammatory markers NLR, PLR, and SII and age-related cataracts via weighted multivariate regression analyses and subgroup analyses. Smoothed curve fitting was performed to identify nonlinear associations and saturation effects between inflammation indices and cataract risk. Finally, receiver operating characteristic (ROC) curves were plotted for factors significantly associated with the development of cataracts to identify the optimal diagnostic inflammation index. This study included 8887 participants without cataracts and 935 participants with cataracts. Multivariate logistic regression analyses after adjusting for covariates revealed that a high SII (OR = 1.000, 95% CI = 1.000–1.000; $P$ = 0.017) and high NLR (OR = 1.065, 95% CI = 1.000–1.134; $P$ = 0.048) were independent risk factors for cataracts. Subgroup analyses did not reveal interactions between the SII, NLR, or cataract and covariates. Smoothed curve fits of the relationships between the SII or NLR and cataracts did not show positive significant saturating effect values for any of the variables. The ROC curve revealed some diagnostic value for cataracts for both the SII (AUC = 0.549, $P$ < 0.001) and the NLR (AUC = 0.603, $P$ < 0.001), but both had weak diagnostic

**Funding:** Chinese medicine inheritance and innovation "Millions upon millions" talent project (Qihuang project) Qihuang scholars (No. 284 of the National Chinese Medicine Education Development [2018]).

**Competing interests:** The authors have declared that no competing interests exist.

value. Our study suggests that the SII and NLR are independent risk factors for cataracts in U.S. adults, but no such associations was identified between the PLR and cataracts.

## Introduction

Although cataract surgery is now well established, cataracts remain one of the most common causes of visual impairment and blindness in middle-aged and elderly people [1–3]. Cataracts significantly reduce the quality of life of patients, e.g., visual impairment alters the quality of daily life, socialization, and cognitive functioning, and they even directly increase the risk of falls, leading to increased mortality [4]. The development of society directly correlates with the public demand for good vision at an early stage. As a result of this demand and an aging population, the need for cataract surgery increases further each year [5]. Even when surgery is performed, intraocular inflammation after cataract surgery is a major problem that needs to be addressed [6]. Thus, cataracts remain a global public health problem.

The search for risk factors for cataracts has been a focus of research in recent years, and early identification and control of risk factors for cataract formation is a difficult and complex problem. Various epidemiological surveys that provide supporting data on cataract risk factors in different regions have been conducted [7–9]. In addition to the recognized aging of the population [10], smoking, alcohol abuse, bright light stimulation, certain systemic metabolic diseases, and ocular diseases may increase the risk of cataracts [8, 11–14].

The opacification of the lens in cataract patients is mainly secondary to inflammation of the intraocular material [15], and chronic systemic inflammatory response causes increased levels of inflammatory factors in the contents of the eye [15, 16]. Venous blood samples from 104 cataract patients and 100 healthy subjects revealed significantly higher levels of interleukin-6 (IL-6), IL-1β, C-reactive protein (CRP) and tumor necrosis factor-1α (TNF-1α) in the cataract group than in the healthy group [17]. In recent years, biomarker indicators calculated from whole-blood parameters have been regarded as an effective strategy to determine the level of systemic inflammation in a simple and rapid manner to improve the efficiency of studies [18]. The neutrophil-to-lymphocyte ratio (NLR), platelet-to-lymphocyte ratio (PLR), and systemic immunoinflammatory index (SII), three low-cost and easily accessible inflammatory marker parameters, are becoming increasingly popular in clinical studies [19] and have been used as important references to evaluate systemic and ocular diseases, such as diabetes [20], pseudo exfoliation syndrome [18] and age-related macular degeneration [21]. Although the NLR, PLR, and SII have excellent predictive value for the risk of other chronic diseases, their relationships with age-related cataracts are currently unknown. Therefore, the aim of our study was to investigate the relationships between the NLR, PLR, and SII calculated from blood indices and cataracts using nationally representative National Health and Nutrition Examination Survey (NHANES) data.

## Materials and methods

### Study population

The data for our study was obtained from the NHANES 2005–2008, a nationally representative cross-sectional study from the United States that includes demographic, socioeconomic, health, and nutritional information. A detailed description of the NHANES database can be found on the following website: https://www.cdc.gov/nchs/nhanes/index.htm. The NHANES included 20,497 participants in 2005–2008 and only administered a questionnaire related to

whether they had had cataract surgery at the age of 20 years and older. Therefore, our study included 9,583 people aged 20 years and older and excluded participants younger than 20 years old. Nine participants who had not explicitly undergone cataract surgery were excluded. A total of 1083 participants who did not have platelets, neutrophils or lymphocytes collected were further excluded. As a result, the study ultimately included 9,822 people to study the correlation between the NLR, PLR, and SII and cataracts, including 935 participants with cataracts and 8,887 participants without cataracts. The participant screening flowchart is shown in Fig 1.

## Primary exposure

The SII, NLR and PLR were used as the main exposure factors in our study. The SII was calculated as the platelet count multiplied by the neutrophil count divided by the lymphocyte count [22]. The NLR is the ratio of the neutrophil count to the lymphocyte count, and the PLR is the ratio of the platelet count to the lymphocyte count. Blood samples were collected at the NHANES Mobile Examination Centers (MECs), and complete blood cell counts were performed via a Beckman Coulter MAXM instrument. The specific methods used for blood collection, processing, and analysis can be found in the NHANES Laboratory/Medical Technician Procedure Manual. Lymphocyte, neutrophil, and platelet counts were recorded at 1000 cells/μL.

## Outcomes

Age-related cataracts were the primary outcome of our study. Because the NHANES does not have a direct diagnosis of cataracts but does have a questionnaire about whether cataract surgery has been performed, the diagnosis of cataracts in terms of outcome largely depended on the questionnaire, which asked adults aged 20 years or older "Have you ever had cataract surgery?". Participants who explicitly answered "yes" or "no" were included in the study, and those who refused to answer, were unsure, or had no response were excluded from the study. Because the visual threshold for cataract surgery has decreased in recent years, regardless of the level of preoperative vision [23], we are more confident that self-reports of cataract surgery are more meaningful for cataract diagnosis. Current NHANES studies on cataract diagnosis have all been based on this questionnaire and have yielded clinical guidance values [24, 25].

## Covariates

A wide range of information on demographic factors and health-related behaviors, such as age, sex, race, education level, poverty income ratio (PIR), marital status, smoking status, alcohol consumption, body mass index (BMI), diabetes mellitus, hypertension, and lipid profiles, was obtained from the NHANES via face-to-face interviews and related examinations. These factors were considered and adjusted for as potential confounders in this study. Race was categorized as Mexican American, non-Hispanic white, non-Hispanic black, or other. Educational attainment was based on points and categorized as below a high school diploma and a high school diploma or more. Marital status was classified as unmarried, married (including living with a partner) or other. BMI was calculated as weight (kg) divided by height squared ($m^2$) and categorized as underweight/normal, overweight, and obese for values of 25.0 kg/m$^2$ and 30.0 kg/m$^2$, respectively. PIR was categorized as <1.5, 1.3–35, or >3.5. Smoking status was analyzed according to never, former, or current smoker. Drinking status was categorized as excessive drinking, moderate drinking, light drinking, or never drinking. Three drinks per day for women and four drinks per day for men were considered excessive drinking. Moderate drinking was defined as two drinks per day for women and three drinks per day for men. Other alcohol consumption was considered light. Diabetes mellitus was defined as (1) fasting blood glucose ≥7.0 mmol/L, (2) 2-hour blood glucose ≥11.1 mmol/L, (3) HbA1c ≥6.5%, and

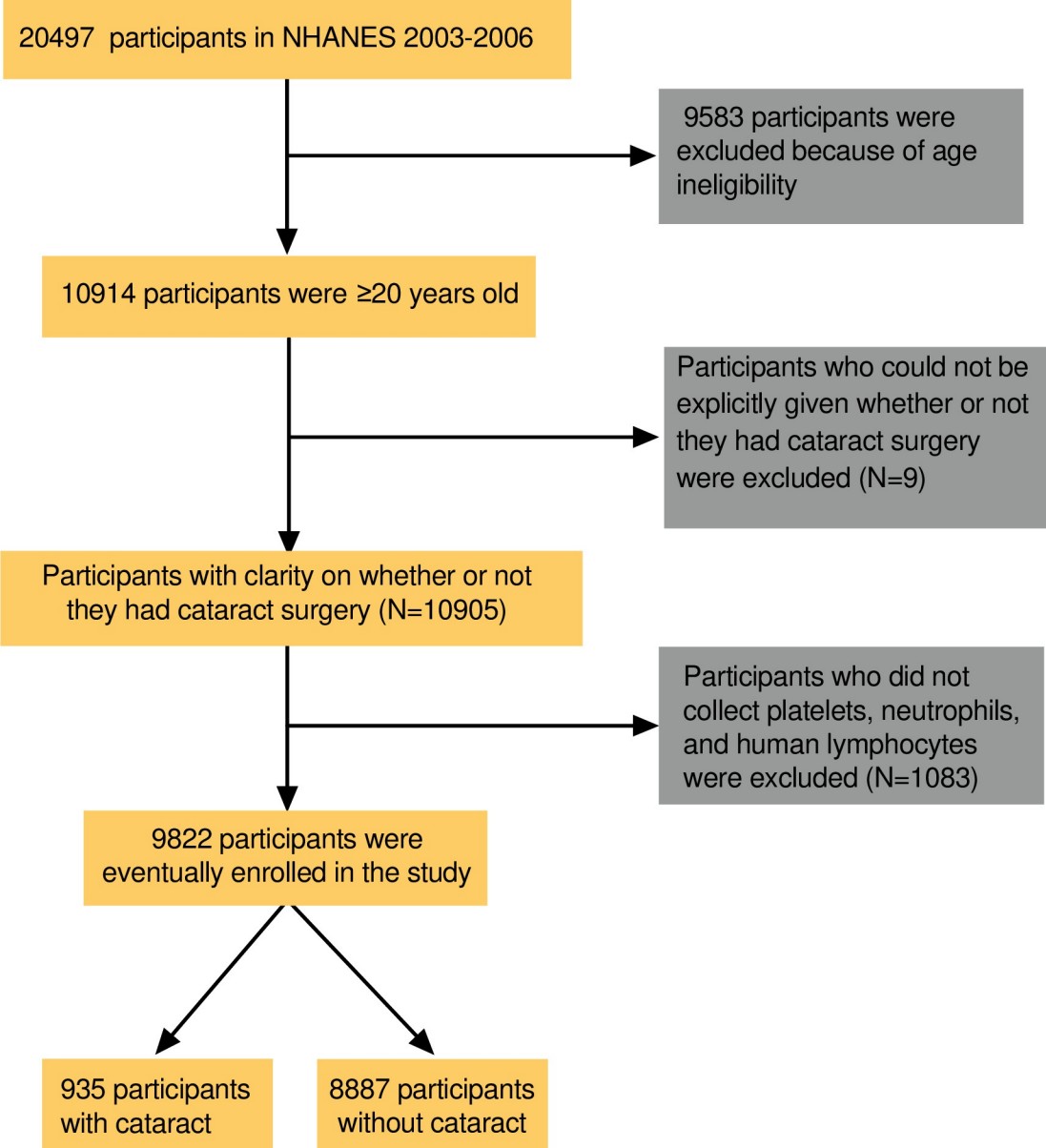

**Fig 1. Flow diagram showing selection of study participants.**

(4) an answer of "yes" to the question "Has a doctor or health professional ever told you that you have diabetes or sugar?". Hypercholesterolemia was defined as a total cholesterol level of at least 6.216 mmol/L or the use of prescription medications to lower cholesterol levels. Hypertension was defined as having a mean systolic blood pressure ≥140 mmHg, a mean diastolic blood pressure ≥90 mmHg, taking prescription medication to lower blood pressure, or having been diagnosed with high blood pressure by a doctor.

## Statistical analysis

Given the complex sampling design of the NHANES, weighted post hoc statistical analyses were used to compare the basic characteristics of participants. Continuous variables with a

normal distribution are described as the mean ± SD, and those not obeying a normal distribution are described as the median (interquartile range, IQR) and were compared between groups via linear regression; categorical variables are expressed as the N (percentage), and the chi-square test was used for between-group comparisons. We constructed binomial logistic regression models to assess whether the NLR, PLR, and SII were risk factors for the development of age-related cataracts, using the presence of cataracts as an outcome indicator. Three models were selected to improve the clarity of associations between variables. Model 1 is the unadjusted model. Model 2 corrects for age, race, and sex. Model 3 builds Model 2 by further correcting for education, marital status, PIR, body mass index, smoking, alcohol consumption, diabetes, hypercholesterolemia, and hypertension. To clarify the relationships between the NLR, PLR, and SII and age-related cataracts, we further performed a 4-quartile regression analysis between the serum NLR, PLR, and SII and cataracts after dividing their values into 4 quartiles. Furthermore, the associations among the NLR, the SII and cataracts, which were statistically significant according to regression analysis, were stratified according to prominent risk factors such as age, race, sex, BMI, diabetes, hyperlipidemia, and hypertension. Smoothed curve fitting was performed to identify nonlinear associations and saturation effects between inflammatory indices and cataract risk. Finally, to identify the optimal diagnostic index among the inflammatory indices, we plotted the receiver operating characteristic (ROC) curves of the indices that were significantly associated with cataracts and calculated the area under the curve (AUC).

All statistical analyses were performed with R software (http://www.R-project.org, The R Foundation, Austria), Empowerstats (http://www.empowerstats.com, X&Y Solutions, Inc., CA, USA), and STATA 16.0 (StataCorp, College Station, TX, USA). The statistical significance level was set at $P < 0.05$.

## Results

### Characteristics of the participants

A total of 8887 participants without cataracts and 935 participants with cataracts who met the screening criteria were included in this study. Notably, we the cohort included cataract patients who were older, unmarried women with cataracts were predominant in the cohort. The risk of cataracts inversely correlated with family income and level of education. Furthermore, the risk of cataracts was greater in individuals with a BMI of less than 30 than in those with a BMI of more than 30. Patients who smoke, consume alcohol, have diabetes, have hypercholesterolemia or hypertension were more likely to develop cataracts. Moreover, participants with cataracts tended to have higher SII, NLR and PLR values. All the baseline profile data are shown in Table 1.

### Relationships between biomarker parameters and cataracts

Regression analyses between SII, NLR, PLR, and cataract are shown in Table 2. After logistic regression analysis, all the models revealed a greater SII as a significant risk factor for cataracts (Model 1: OR = 1.000, 95% CI = 1.000–1.001, $P<0.001$; Model 2: OR = 1.000, 95% CI = 1.000–1.000, $P = 0.004$; Model 3: OR = 1.000, 95% CI = 1.000–1.000, $P = 0.017$). To better clarify the relationship between the SII and cataracts, we further investigated the associations between the SII values of different grades and cataracts after the SII values were divided into quartiles. In all 3 models, cataract incidence was greater in Q4 than in Q1 of the SII population (Model 1: OR = 1.579, 95% CI = 1.308–1.906, $P<0.001$; Model 2: OR = 1.406, 95% CI = 1.117–1.771, $P = 0.004$; Model 3: OR = 1.392, 95% CI = 1.102–1.759, $P = 0.006$). The P values indicated significant differences in all models (Model 1: $P<0.001$; Model 2: $P = 0.001$; Model 3: $P = 0.002$).

**Table 1. Basic participant characteristics.**

| Variables | Without Cataract | Cataract | *P* value |
|---|---|---|---|
| No. of subjects | 8887 | 935 | |
| Age, years | 46.00 (33.00–61.00) | 78.00 (70.00–80.00) | **<0.001** |
| Sex, % | | | **<0.001** |
| Male | 4340 (48.84%) | 419 (44.81%) | |
| Female | 4547 (51.17%) | 516 (55.19%) | |
| Race, % | | | **<0.001** |
| Mexican American | 1779 (20.02%) | 72 (7.70%) | |
| Non-Hispanic White | 4161 (46.82%) | 660 (70.59%) | |
| Non-Hispanic Black | 1903 (21.41%) | 118 (12.62%) | |
| Other Race | 1044 (11.75%) | 85 (9.09%) | |
| Family PIR, % | | | **<0.001** |
| <1.5 | 2825 (31.79%) | 304 (32.51%) | |
| 1.5–3.5 | 2697 (30.35%) | 373 (39.89%) | |
| >3.5 | 2782 (31.30%) | 174 (18.61%) | |
| Not recorded | 583 (6.56%) | 84 (8.98%) | |
| Education Level, % | | | **<0.001** |
| Less than high school | 2528 (28.45%) | 344 (36.79%) | |
| High school or equivalent | 2133 (24.00%) | 246 (26.31%) | |
| More than high school | 4218 (47.46%) | 344 (36.79%) | |
| Not recorded | 8 (0.09%) | 1 (0.11%) | |
| Marital status, % | | | **<0.001** |
| Married or living with a partner | 5596 (62.97%) | 474 (50.69%) | |
| Unmarried or other | 3288 (37.00%) | 460 (49.20%) | |
| Not recorded | 3 (0.03%) | 1 (0.11%) | |
| BMI, kg/m$^2$, % | | | **<0.001** |
| <25 | 2570 (28.92%) | 288 (30.80%) | |
| 25–30 | 3006 (33.83%) | 328 (35.08%) | |
| ≥30 | 3191 (35.91%) | 282 (30.16%) | |
| Not recorded | 120 (1.35%) | 37 (3.96%) | |
| Smoking, % | | | **<0.001** |
| Never | 4701 (52.90%) | 453 (48.45%) | |
| Former | 2094 (23.56%) | 395 (42.25%) | |
| Current | 2086 (23.47%) | 85 (9.09%) | |
| Not recorded | 6 (0.07%) | 2 (0.21%) | |
| Drinking status, % | | | **<0.001** |
| Never | 1142 (12.85%) | 194 (20.75%) | |
| Light alcohol consumption | 3524 (39.65%) | 487 (52.09%) | |
| Moderate alcohol consumption | 1098 (12.36%) | 58 (6.20%) | |
| Excessive alcohol consumption | 2518 (28.33%) | 124 (13.26%) | |
| Not recorded | 605 (6.81%) | 72 (7.70%) | |
| Diabetes, % | | | **<0.001** |
| Yes | 1361 (15.32%) | 331 (35.40%) | |
| No | 7513 (84.54%) | 603 (64.49%) | |
| Hypertension, % | | | **<0.001** |
| Yes | 7837 (88.19%) | 878 (93.90%) | |
| No | 794 (8.93%) | 37 (3.96%) | |
| Not recorded | 256 (2.88%) | 20 (2.14%) | |

(*Continued*)

**Table 1.** (Continued)

| Variables | Without Cataract | Cataract | *P* value |
|---|---|---|---|
| Hypercholesterolemia, % | | | **<0.001** |
| Yes | 2433 (27.38%) | 432 (46.20%) | |
| No | 6454 (72.62%) | 503 (53.80%) | |
| SII | 517.00 (369.75–732.65) | 564.57 (395.60–833.39) | **<0.001** |
| NLR | 1.95 (1.46–2.61) | 2.31 (1.70–3.08) | **<0.001** |
| PLR | 127.50 (101.90–162.05) | 136.25 (106.15–178.50) | **<0.001** |

A bold P value indicates a statistically significant difference. PIR, poverty income ratio; BMI, body mass index; SII, systemic immune–inflammation index; NLR, neutrophil–lymphocyte ratio; PLR, platelet–lymphocyte ratio.

For the total NLR, all the models identified a high NLR as a significant risk factor for cataracts (Model 1: OR = 1.255, OR = 1.199–1.313, $P<0.001$; Model 2: OR = 1.076, 95% CI = 1.014–1.143, $P = 0.016$; Model 3: OR = 1.065, 95% CI = 1.000–1.134, $P = 0.048$). The NLR was categorized into quartiles to investigate the associations between different NLR values and cataracts. In all 3 models, cataract incidence was greater in Q4 than in Q1 of the NLR

**Table 2.** Associations between the SII and cataract incidence in different quintiles.

| Variables | Model 1 | Model 2 | Model 3 |
|---|---|---|---|
| | OR (95% CI) *P* value | OR (95% CI) *P* value | OR (95% CI) *P* value |
| SII | **1.000 (1.000–1.001) <0.001** | **1.000 (1.000–1.000) 0.004** | **1.000 (1.000–1.000) 0.017** |
| NLR | **1.255 (1.199–1.313) <0.001** | **1.076 (1.014–1.143) 0.016** | **1.065 (1.000–1.134) 0.048** |
| PLR | **1.004 (1.003–1.005) <0.001** | 1.001 (0.999–1.002) 0.301 | 1.001 (0.999–1.002) 0.357 |
| SII quartiles | | | |
| < 371.95 | Reference | Reference | Reference |
| 371.95–521.83 | 1.047 (0.855–1.282) 0.658 | 1.039 (0.818–1.320) 0.754 | 1.043 (0.818–1.328) 0.736 |
| 521.83–740.94 | 1.132 (0.927–1.381) 0.225 | 1.130 (0.890–1.435) 0.316 | 1.114 (0.874–1.419) 0.384 |
| > 740.94 | **1.579 (1.308–1.906) <0.001** | **1.406 (1.117–1.771) 0.004** | **1.392 (1.102–1.759) 0.006** |
| *p* for trend | **<0.001** | **0.001** | **0.002** |
| NLR quartiles | | | |
| < 1.47 | 1.0 | 1.0 | 1.0 |
| 1.47–1.98 | **1.462 (1.171–1.826) <0.001** | 1.289 (0.993–1.672) 0.056 | 1.274 (0.975–1.664) 0.076 |
| 1.98–2.66 | **1.699 (1.370–2.108) <0.001** | 1.259 (0.975–1.626) 0.078 | 1.207 (0.927–1.571) 0.162 |
| > 2.66 | **2.709 (2.211–3.319) <0.001** | **1.527 (1.194–1.952) <0.001** | **1.428 (1.107–1.842) 0.006** |
| *p* for trend | **<0.001** | **0.002** | **0.014** |
| PLR quartiles | | | |
| < 102.07 | 1.0 | 1.0 | 1.0 |
| 102.07–128.20 | 0.943 (0.771–1.154) 0.569 | 1.010 (0.795–1.284) 0.933 | 1.020 (0.796–1.306) 0.878 |
| 128.20–163.71 | 1.021 (0.837–1.245) 0.840 | 1.115 (0.879–1.413) 0.371 | 1.158 (0.907–1.480) 0.240 |
| 163.71–628.57 | **1.503 (1.249–1.808) <0.001** | 1.114 (0.891–1.392) 0.344 | 1.101 (0.872–1.390) 0.417 |
| *p* for trend | **<0.001** | 0.280 | 0.354 |

Bold *P* values indicate statistically significant differences.

Model 1: no adjustment for covariates.

Model 2: adjusted for age, race, and sex.

Model 3: further adjusted for education, marital status, the household poverty-to-income ratio, body mass index, smoking, alcohol consumption, diabetes, hypercholesterolemia, and hypertension.

population (Model 1: OR = 2.709, 95% CI = 2.211–3.319, $P<0.001$; Model 2: OR = 1.527, 95% CI = 1.194–1.952, $P<0.001$; Model 3: OR = 1.428, 95% CI = 1.107–1.842, $P = 0.006$). The P values indicated significant differences in all models (Model 1: $P<0.001$; Model 2: $P = 0.002$; Model 3: $P = 0.014$).

Model 1, which was not adjusted for any covariates, revealed that a high PLR was a significant risk factor for cataracts (Model 1: OR = 1.004, OR = 1.003–1.005, $P<0.001$), whereas neither Model 2 nor Model 3 showed an association between the PLR and cataracts. Quartile stratification studies in which the PLR was corrected for covariates also did not reveal significant associations.

## Subgroup analysis

Table 3 shows the associations between SII, NLR, and cataracts further stratified by age, sex, race, BMI, diabetes status, hyperlipidemia status, and hypertension status. After we corrected for variables other than subgroup variables, we observed that the associations between the SII or NLR and cataracts were not affected by covariate interactions ($P$ for interaction > 0.05).

**Table 3. Associations between the SII or NLR and cataracts according to age, race, sex, BMI, diabetes, hyperlipidemia status, and hypertension status.**

| Variables | SII | | NLR | |
|---|---|---|---|---|
| | **Model: OR (95% CI)** | *P* for interaction | **Model: OR (95% CI)** | *P* for interaction |
| | *P* value | | *P* value | |
| Age (years) | | 0.852 | | 0.644 |
| 20–59 | 1.000 (1.000–1.001) 0.366 | | 1.074 (0.869–1.329) 0.509 | |
| 60–85 | **1.000 (1.000–1.001) 0.001** | | **1.130 (1.062–1.204) <0.001** | |
| Sex, % | | 0.061 | | 0.066 |
| Male | 1.000 (1.000–1.000) 0.479 | | 1.011 (0.930–1.100) 0.797 | |
| Female | **1.000 (1.000–1.001) 0.001** | | **1.135 (1.035–1.246) 0.007** | |
| Race, % | | 0.069 | | 0.073 |
| Mexican American | 1.000 (0.999–1.001) 0.738 | | 0.913 (0.721–1.157) 0.453 | |
| Non-Hispanic White | **1.000 (1.000–1.001) 0.005** | | 1.065 (0.990–1.145) 0.090 | |
| Non-Hispanic Black | 1.000 (0.999–1.000) 0.475 | | 1.006 (0.829–1.221) 0.951 | |
| Other Race | **1.001 (1.000–1.002) 0.008** | | **1.391 (1.096–1.766) 0.007** | |
| BMI, % | | 0.404 | | 0.290 |
| <25 | 1.000 (1.000–1.001) 0.225 | | 1.041 (0.941–1.152) 0.436 | |
| 25–30 | 1.000 (1.000–1.000) 0.526 | | 1.025 (0.923–1.138) 0.640 | |
| ≥30 | **1.000 (1.000–1.001) 0.013** | | **1.150 (1.025–1.290) 0.017** | |
| Diabetes, % | | 0.493 | | 0.534 |
| Yes | **1.000 (1.000–1.001) 0.045** | | 1.098 (0.979–1.231) 0.109 | |
| No | 1.000 (1.000–1.000) 0.137 | | 1.052 (0.978–1.133) 0.175 | |
| Hypertension, % | | 0.302 | | 0.272 |
| Yes | **1.000 (1.000–1.001) 0.022** | | **1.108 (1.008–1.218) 0.033** | |
| No | 1.000 (1.000–1.000) 0.187 | | 1.035 (0.954–1.122) 0.411 | |
| Hypercholesterolemia, % | | 0.247 | | 0.719 |
| Yes | 1.000 (1.000–1.000) 0.577 | | 1.051 (0.953–1.159) 0.323 | |
| No | **1.000 (1.000–1.001) 0.010** | | 1.075 (0.992–1.165) 0.077 | |

Bold *P* values indicate statistically significant differences.

Model: adjusted for age, race, gender education, marital status, household poverty-to-income ratio, body mass index, smoking, alcohol consumption, diabetes, hypercholesterolemia, and hypertension variables, but the model was not adjusted for the stratification variables themselves.

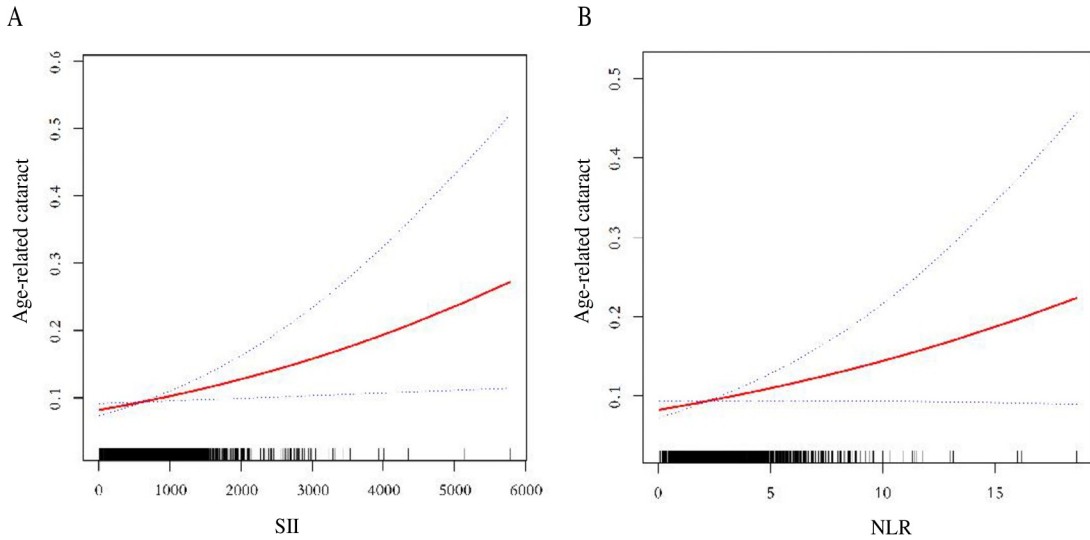

**Fig 2. Relationship between SII, NLR and cataract risk.** The red solid line represents a smoothed curve fit of SII to cataract risk. The blue dashed line represents the 95% confidence interval of the smoothed curve fit. (A) Relationship between SII and cataract risk; (B) Relationship between NLR and cataract risk.

## Nonlinearity and saturation effect analysis between the SII and NLR and cataract incidence

We further implemented smooth curve fitting of the SII, NLR, and cataract relationship to identify nonlinear associations and saturation effects (Fig 2). A saturation effect value of 552 was found for the total SII-cataract relationship, but this saturation effect was not significant (Table 4). A saturation effect value of 2.769 was identified for the total NLR-cataract relationship, but this saturation effect also was not significant (Table 4).

## ROC curves for cataract diagnosis by the SII and NLR

The SII and NLR, which significantly correlated with cataract risk in the corrected multivariate regression model, were plotted on ROC curves to calculate the predictive effect of inflammatory markers on cataracts. As shown in Fig 3, the AUC = 0.549, $P < 0.001$ for the SII and AUC = 0.603, $P < 0.001$ for the NLR. Both the NLR and the SII showed weak diagnostic value

**Table 4. Saturation effect analysis of the SII on age-related cataracts.**

| Variables | Model: saturation effect analysis [β (95% CI) $p$ value] |
|---|---|
| SII | |
| turning point (K) | 552 |
| <K, effect1 | 1.001 (1.000–1.001) 0.173 |
| >K, effect2 | 1.000 (1.000–1.000) 0.154 |
| Log-likelihood ratio | 0.449 |
| NLR | |
| turning point (K) | 2.769 |
| <K, effect1 | 1.146 (0.948–1.386) 0.160 |
| >K, effect2 | 0.980 (0.891–1.078) 0.679 |
| Log-likelihood ratio | 0.070 |

## ROC curve for AGE.RELATED.CATARACT

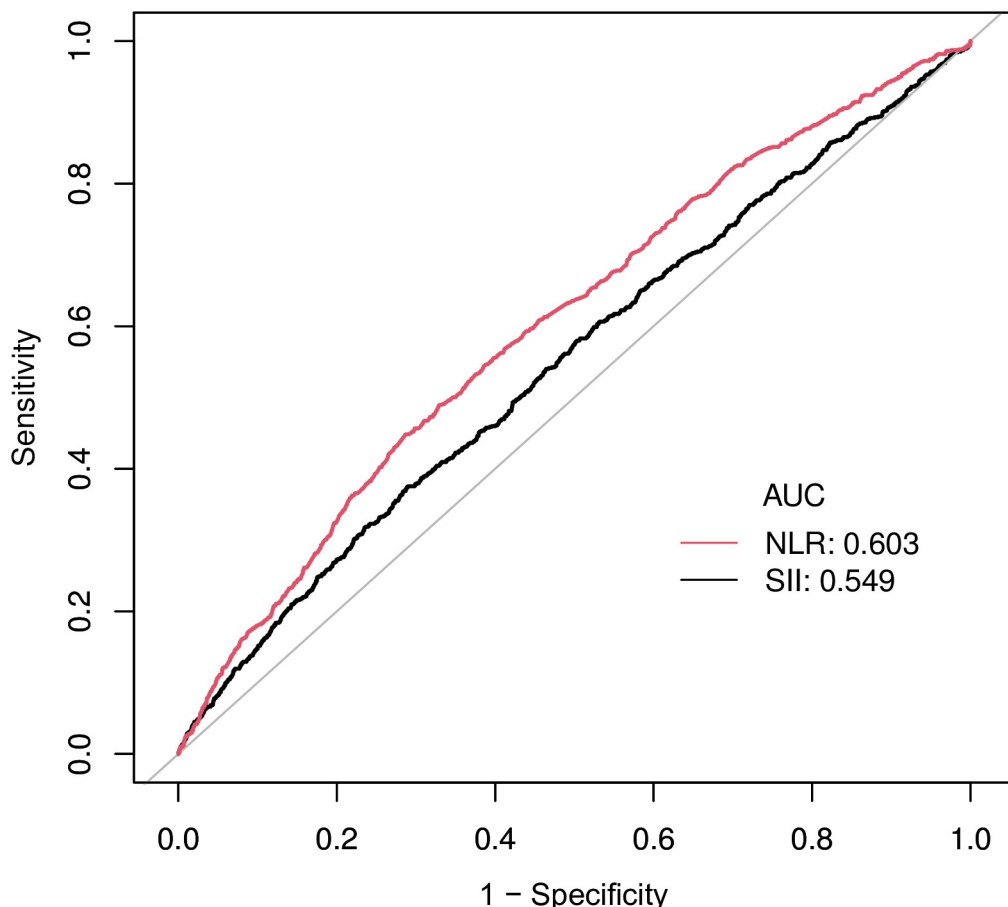

**Fig 3. ROC analysis of SII, NLR for cataract diagnosis.** AUC, area under curve.

for cataracts because the AUCs were all less than 0.7; therefore, no further comparisons are necessary.

## Discussion

This work is the first comprehensive study of the associations between the peripheral blood inflammatory markers SII, PLR, and NLR and age-related cataracts. The serum SII and NLR were found to be significantly and positively associated with the incidence of age-related cataracts, independent of traditionally considered correlates, including age, sex, race, household income, BMI, alcohol consumption, smoking status, and systemic metabolic diseases. However, we did not identify a correlation between the PLR and cataract incidence.

Although inflammation is usually viewed as a localized or systemic event, chronic inflammation, which is characterized by persistent low-level infiltration of leukocytes in localized tissues, is so prevalent in the aging process that the term "inflammatory senescence" has been used to describe this state [26]. Age-related cataracts are the result of persistent lens "inflammatory senescence" due to various chronic causes. Inflammation is indeed a key factor in the development of cataracts, as evidenced by the increased levels of IL-1, IL-1β, IL-6, and TNF-alpha in the vitreous fluid of cataract patients [27, 28].

The leukocyte count and its many subtypes are used to identify inflammation in the clinic [29]. Neutrophils, the major leukocyte sublineage, are dramatically recruited to inflamed tissues in the early acute inflammatory response to combat and clear invading pathogens [30]. Neutrophils can exacerbate inflammation in ocular tissues by releasing several proinflammatory mediators [31], whereas lymphocytes play the opposite role: they primarily modulate the inflammatory response [32]; for example, lymphocytes have an antiatherosclerotic effect [33]. Platelets have traditionally been recognized as an important link in hemostasis, but recent studies have also clarified the prominent contribution of platelets to the inflammatory process. The expression of these adhesion molecules and platelet surface receptors, such as platelet surface integrins, complement receptors, and the Toll-like receptor family, provides evidential support for the stimulatory and regulatory role of platelets in inflammatory stimuli and immune responses [34]. Platelets in the physiological state appear to circulate in the bloodstream in a resting state, and platelets are activated by aging-related inflammation and interact with leukocytes circulating in the bloodstream [35]. Platelets mediate the inflammatory response to leukocyte migration and exudation by releasing VEGF and platelet-derived growth factor (PDGF) [33, 36].

Originally developed to reflect the intensity of systemic inflammation and stress in critically ill patients, the peripheral blood inflammation index has since been shown to have predictive value in the acute and chronic inflammatory responses of various diseases and has received widespread attention from clinical practitioners because of its ease of accessibility, reproducibility, and inexpensiveness [37]. Various inflammatory markers derived from absolute counts of neutrophils, lymphocytes, and platelets have been used in a variety of disease prognoses, disease recurrences, and responses to therapy [33]. The NLR and PLR are highly valuable for assessing the severity of systemic inflammation and predicting inflammatory infections and other comorbidities. The SII, an immunoinflammatory evaluation index that integrates the NLR and PLR, is more representative. However, our study did not identify an association between the PLR and cataracts, whereas the SII and NLR provide potential, albeit weaker, diagnostic value for cataracts. The present study focused on the NLR and the SII as risk factors for cataracts. Because of the high prevalence of cataracts, the importance of the NLR and the SII is to provide clinicians with new ideas, especially in the early prevention and intervention of cataracts, which are still monitored by specialized ophthalmic equipment. Another important contribution of this finding is that it strengthens the association between ophthalmology and internal medicine and promotes the integration of these disciplines.

For the first time, we investigated the associations between peripheral blood inflammatory markers and cataracts, and this cross-sectional study has strengths and limitations. The strengths are the large sample size and the covariate corrections that were considered as comprehensively as possible, which enhanced the reliability and representativeness of this study. However, our study also has limitations, mainly in the control of confounders, the extraction of endpoints, and the shortcomings of the cross-sectional study itself. First, we were unable to obtain useful data on the confounding factors that cause cataracts, such as light stimulation, genetic factors, and steroid hormone levels in the body, because these data were not included in the NHANES [38]. Second, our diagnosis of cataract was based on whether the patient had undergone cataract surgery, which invariably narrows the screening for cataracts. Moreover, some patients who already have cataracts may have consequently been included in the control group. Because we lacked information on the degree of lens clouding, the severity of cataracts could not be graded, which precluded a detailed stratification of the diagnostic value of risk factors for cataracts. Finally, cross-sectional studies cannot provide a clear causal relationship and cannot avoid reverse causation. The relationship between inflammation and cataracts is

complex, and our study is motivated mainly by convenient clinical diagnostic considerations; specific mechanisms still need to be explored via deeper targeted experiments.

## Conclusions

Recent studies have demonstrated that the SII, NLR and PLR, which are based on peripheral lymphocyte, neutrophil and platelet counts, reflect local and systemic immune-inflammatory responses and have high diagnostic value for the prognosis of a wide range of clinical conditions. This study primarily focused on the value of the SII, NLR and PLR in the diagnosis and risk estimation of cataract patients. Multivariate regression analysis revealed that the SII and NLR may be risk factors for cataracts, whereas the PLR is not associated with the risk of cataracts. Despite the limitations of our study, this study is valuable because it is a relatively comprehensive cross-sectional study, suggesting that the NLR and the SII may be potential predictors of inflammation in cataract patients because of their low-cost and easily accessible blood indices.

## Acknowledgments

Thanks to all the participants and staff of the NHANES for their tremendous contributions in data collection, management, and publication.

## Author Contributions

**Conceptualization:** Baohua Li, Zefeng Kang.

**Data curation:** Baohua Li, Xinyue Hou, MingMing Zhang.

**Formal analysis:** Baohua Li, Zefeng Kang.

**Methodology:** Baohua Li, Xinyue Hou, Zefeng Kang.

**Project administration:** Xinyue Hou.

**Resources:** Xinyue Hou.

**Software:** Baohua Li, Xinyue Hou, Bobiao Ning.

**Supervision:** Baohua Li, Bobiao Ning, Xiao Li, MingMing Zhang, Jianquan Wang, Mengyu Liu, Yipeng Shi, Zefeng Kang.

**Validation:** Bobiao Ning, MingMing Zhang, Jianquan Wang, Mengyu Liu, Yipeng Shi, Zefeng Kang.

**Visualization:** Xiao Li, Zefeng Kang.

**Writing – original draft:** Baohua Li, Bobiao Ning.

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
