## [Decision Letter · Decision Letter 0]

15 Aug 2024

PONE-D-24-25660Predictive role of peripheral blood inflammation indices NLR, PLR, and SII for age-related cataract riskPLOS ONE

Dear Dr. Kang,

Thank you for submitting your manuscript to PLOS ONE. After careful consideration, we feel that it has merit but does not fully meet PLOS ONE’s publication criteria as it currently stands. Therefore, we invite you to submit a revised version of the manuscript that addresses the points raised during the review process.

We look forward to receiving your revised manuscript.

Kind regards,

Aleksandra Klisic

Academic Editor

PLOS ONE

2. PLOS requires an ORCID iD for the corresponding author in Editorial Manager on papers submitted after December 6th, 2016. Please ensure that you have an ORCID iD and that it is validated in Editorial Manager. To do this, go to ‘Update my Information’ (in the upper left-hand corner of the main menu), and click on the Fetch/Validate link next to the ORCID field. This will take you to the ORCID site and allow you to create a new iD or authenticate a pre-existing iD in Editorial Manager. Please see the following video for instructions on linking an ORCID iD to your Editorial Manager account: https://www.youtube.com/watch?v=_xcclfuvtxQ".

Reviewers' comments:

Reviewer's Responses to Questions

**Comments to the Author**

1. Is the manuscript technically sound, and do the data support the conclusions?

Reviewer #1: Yes

Reviewer #2: Yes

Reviewer #3: Partly

2. Has the statistical analysis been performed appropriately and rigorously? 

Reviewer #1: No

Reviewer #2: Yes

Reviewer #3: Yes

3. Have the authors made all data underlying the findings in their manuscript fully available?

Reviewer #1: Yes

Reviewer #2: No

Reviewer #3: Yes

4. Is the manuscript presented in an intelligible fashion and written in standard English?

Reviewer #1: Yes

Reviewer #2: Yes

Reviewer #3: Yes

5. Review Comments to the Author

Reviewer #1: General Comments:

This is an interesting analysis of the possible relationship between NLR, PLR, SII and age-related cataract. The experimental design appears good, but the interpretation of the results is highly debatable, as well as the potential clinical utility of these findings to ophthalmologists. The manuscript could benefit significantly also from the statement of a clear hypothesis to be tested in the introduction section, to better clarify to readers the intent and relevance of the studies. Finally the manuscript would also benefit from editing by a primary English-speaking editor.

Specific comments

Comment 1: The introduction section lacks any information about the inflammatory markers that have been studied in the pathogenesis of cataract.

Comment 2: The introduction section does not contain the study's aims.

Comment 3: The authors defined dyslipidemia as a total cholesterol level of at least 6.216 mmol/L or the use of prescription medications to lower cholesterol levels. Dyslipidemia is a much broader term and the criteria of the author correspond to hypercholesterolemia.

Comment 4: All continuous variables are presented as mean ± SD. This is correct only for variables with normal distribution. Does PRL values normally distributed?

Comment 5: Categorical variables with two labels (yes/no) are not adequately presented in the Table 1.

Comment 6: Interpretation of the ROC curves analysis for cataract diagnosis by SII, NLR is incorrect. In general, an AUC of 0.5 suggests no discrimination, 0.7 to 0.8 is considered acceptable, 0.8 to 0.9 is considered excellent, and more than 0.9 is considered outstanding. According with the authors results it is really unacceptable to use these parameters for controls and cataract patient’s discrimination.

Comment 7: In most pathological conditions, there is an increase in markers of inflammation. How cataract is a disease of elderly people who in most cases have another disease what are the clinical significance of the whole study when the parameters are not specific for cataract?

Reviewer #2: Li et al. have performed a study on the predictive role of NLR, PLR, and SII for age-related cataract risk. The study findings are interesting and the manuscript is well-written. These are my comments:

- The title should be free of abbreviations if the word limit allows.

- A paragraph summarizing the clinical take-home message of this manuscript should be added to the discussion.

- The references prior to 2010 could be updated with those after 2010 since they provide more up-to-date findings.

Reviewer #3: In this manuscript, the authors compared the NLR, SII, and PLR levels of 935 cataract patients registered in the NHANES database with a population of 8887 participants.

1. The purpose of the study is not clear. Why did the authors investigate NLR, SII, and PLR in cataract patients in this study?

2. Did the cataract group consist of those who had undergone cataract surgery, or were both those who had undergone cataract surgery and those who had not? If the participants had already undergone cataract surgery, what was the purpose of assessing NLR, SII, and PLR in these individuals, and what would this do?

3. How do the authors explain the lower risk of cataract in those with BMI <30?

4. The authors should provide insight into how NLR and SII can be used in clinical practice for cataract. Should everyone with elevated levels of these parameters be screened for cataract? If so, these parameters will increase in numerous chronic and acute inflammatory conditions and are not specific to cataracts. The authors consider how to overcome this handicap. In addition, older age is more associated with cataract risk than these parameters, and if they are to be used for cataract screening, it seems more cost-effective to screen the elderly population. If they are recommended for diagnosis, it is impossible to diagnose cataract by evaluating these parameters. Because increases in these parameters can be detected in many people due to many reasons.

5. Ratios such as NLR and SII are calculated from data obtained from complete blood counts and are continuously variable values that are affected by many conditions and very quickly. As the authors also mentioned, examining these parameters only once in a chronic pathological condition such as cataract is devoid of establishing a cause-effect relationship.

6. The English language contains many grammatical and typo errors.

6. PLOS authors have the option to publish the peer review history of their article (what does this mean?). If published, this will include your full peer review and any attached files.

Reviewer #1: No

Reviewer #2: No

Reviewer #3: No

---

## [Author Response · Author response to Decision Letter 0]

4 Oct 2024

We have revised it according to the requirements and if there are still irregularities please give us another chance to revise.

2. PLOS requires an ORCID iD for the corresponding author in Editorial Manager on papers submitted after December 6th, 2016. Please ensure that you have an ORCID iD and that it is validated in Editorial Manager. To do this, go to ‘Update my Information’ (in the upper left-hand corner of the main menu), and click on the Fetch/Validate link next to the ORCID field. This will take you to the ORCID site and allow you to create a new iD or authenticate a pre-existing iD in Editorial Manager. Please see the following video for instructions on linking an ORCID iD to your Editorial Manager account: https://www.youtube.com/watch?v=_xcclfuvtxQ".

The ORCID iD has been validated as required.

Reviewers' comments:

Reviewer's Responses to Questions

Comments to the Author

1. Is the manuscript technically sound, and do the data support the conclusions?

Reviewer #1: Yes

Reviewer #2: Yes

Reviewer #3: Partly

2. Has the statistical analysis been performed appropriately and rigorously? 

Reviewer #1: No

Reviewer #2: Yes

Reviewer #3: Yes

3. Have the authors made all data underlying the findings in their manuscript fully available?

Reviewer #1: Yes

Reviewer #2: No

Reviewer #3: Yes

4. Is the manuscript presented in an intelligible fashion and written in standard English?

Reviewer #1: Yes

Reviewer #2: Yes

Reviewer #3: Yes

5. Review Comments to the Author

Please use the space provided to explain your answers to the questions above. You may also include additional comments for the author, including concerns about dual publication, research ethics, or publication ethics. (Please upload your review as an attachment if it exceeds 20,000 characters)。

Reviewer #1: General Comments:

This is an interesting analysis of the possible relationship between NLR, PLR, SII and age-related cataract. The experimental design appears good, but the interpretation of the results is highly debatable, as well as the potential clinical utility of these findings to ophthalmologists. The manuscript could benefit significantly also from the statement of a clear hypothesis to be tested in the introduction section, to better clarify to readers the intent and relevance of the studies. Finally the manuscript would also benefit from editing by a primary English-speaking editor.

We have taken your suggestion to present an explicit hypothesis to be tested in the introduction section to better clarify the intention and relevance of the study to the reader.

Sorry, we do need to improve on the language and this revision has been touched up by a professional.

Specific comments

Comment 1: The introduction section lacks any information about the inflammatory markers that have been studied in the pathogenesis of cataract.

We have added further research information on inflammatory markers in cataract pathogenesis in the Introduction, unfortunately there is not much supporting literature. It is added as follows:

A venous blood sample from 104 cataract patients and 100 healthy subjects showed significantly higher levels of interleukin-6 (IL-6), IL-1β, C-reactive protein (CRP) and tumor necrosis factor-1α (TNF-1α) in the cataract group than in the healthy group [PMID : 31081067].

Comment 2: The introduction section does not contain the study's aims. 

Thanks for careful reading, we have added the purpose of the study in the citation.

Although NLR, PLR, and SII provide excellent predictive value for the risk of other chronic diseases, the relationship between them and age-related cataract is currently unknown. Therefore, the aim of our study was to investigate the relationship between NLR, PLR, SII calculated from blood indices and cataract using nationally representative National Health and Nutrition Examination Survey (NHANES) data.

Comment 3: The authors defined dyslipidemia as a total cholesterol level of at least 6.216 mmol/L or the use of prescription medications to lower cholesterol levels. Dyslipidemia is a much broader term and the criteria of the author correspond to hypercholesterolemia. 

Your suggestion is more precise, we have changed hyperlipidaemia to hypercholesterolaemia in the main text.

Comment 4: All continuous variables are presented as mean ± SD. This is correct only for variables with normal distribution. Does PRL values normally distributed? 

Your question reminded us to recalculate normality for continuous variables and make data presentation changes.

Continuous variables were subjected to normality recommendations, those obeying normal distribution were described as mean ± SD and those not obeying normal distribution were described as Median (interquartile range, IQR) and linear regression was used for comparisons between groups.

Comment 5: Categorical variables with two labels (yes/no) are not adequately presented in the Table 1. 

As paraphrased from a question above, the categorical variable is denoted as N (percentage), and I hope we have understood your wishes.

Comment 6: Interpretation of the ROC curves analysis for cataract diagnosis by SII, NLR is incorrect. In general, an AUC of 0.5 suggests no discrimination, 0.7 to 0.8 is considered acceptable, 0.8 to 0.9 is considered excellent, and more than 0.9 is considered outstanding. According with the authors results it is really unacceptable to use these parameters for controls and cataract patient’s discrimination. 

Your comment is correct. Originally, we only wanted to use the AUC value to compare the diagnostic ability of SII and NLR for cataract, but now that the AUC seems to be relatively low, we can only assume that the diagnostic value of the two for cataracts is weaker, and no further comparison will be made.

Comment 7: In most pathological conditions, there is an increase in markers of inflammation. How cataract is a disease of elderly people who in most cases have another disease what are the clinical significance of the whole study when the parameters are not specific for cataract? 

We have considered all of the issues you have raised and have taken into account age, gender, ethnicity, education level, Poverty income ratio (PIR), marital status, smoking status, alcohol consumption, body mass index (BMI), diabetes, hypertension, and lipids as much as possible and corrected for these variables by including them in the regression analyses. Interference by other factors in the results of the study has been avoided as much as possible.

In recent years, biomarker indices calculated from whole blood parameters have been considered as an effective strategy to determine the level of systemic inflammation in a simple and rapid manner in order to improve the efficiency of the study.NLR, PLR and SII, three low-cost and easily accessible inflammation marker parameters, have been used as important reference indicators for the evaluation of systemic and ocular diseases, such as diabetes mellitus.

Reviewer #2: Li et al. have performed a study on the predictive role of NLR, PLR, and SII for age-related cataract risk. The study findings are interesting and the manuscript is well-written. These are my comments:

- The title should be free of abbreviations if the word limit allows.

Your suggestion has been taken on board.

Predictive role of peripheral blood inflammation indices neutrophil-to-lymphocyte ratio (NLR), platelet-to-lymphocyte ratio (PLR), and systemic immunoinflammatory index (SII) for age-related cataract risk

- A paragraph summarizing the clinical take-home message of this manuscript should be added to the discussion. 

The discussion section has been supplemented.

Originally developed to reflect the intensity of systemic inflammation and stress in critically ill patients, the Peripheral Blood Inflammation Index (PLI) has since proven to be of predictive value in acute and chronic inflammatory responses in a variety of diseases, and is of widespread interest to clinical practitioners because of its ease of access, reproducibility, and inexpensiveness.NLR and PLR are valuable in assessing the severity of systemic inflammation and predicting inflammatory infections and other comorbidities. SII is an immunoinflammatory assessment index that integrates NLR and PLR and is more representative. However, our study did not find an association between PLR and cataract, whereas SII and NLR provide potential, albeit weaker, diagnostic value for cataract. The present study focused on NLR and SII as risk factors for cataract. Because of the high prevalence of cataract, the significance of NLR and SII is to provide clinicians with new ideas, mainly in the early prevention and intervention of cataract, which is still monitored by specialised ophthalmic equipment. Another important significance is that it strengthens the association between ophthalmology and internal medicine and promotes the integration of the disciplines.

- The references prior to 2010 could be updated with those after 2010 since they provide more up-to-date findings. 

We have tried our best to remove the literature before 2010 and the references have been updated. However, the literature of has not been updated because it strongly supports the increase of inflammatory factors such as IL-1, IL-1β, and IL-6 in the lens epithelium of cataract patients.

Reviewer #3: In this manuscript, the authors compared the NLR, SII, and PLR levels of 935 cataract patients registered in the NHANES database with a population of 8887 participants.

1. The purpose of the study is not clear. Why did the authors investigate NLR, SII, and PLR in cataract patients in this study? 

We apologize for the lack of a description of the purpose of the study in the introduction, which we have added.

Biomarker indices calculated from whole blood parameters are regarded as an effective strategy to determine the level of systemic inflammation in a simple and rapid manner.NLR, SII, and PLR, three low-cost and readily available inflammatory marker parameters, are becoming increasingly popular in clinical studies and have been used as important references for the evaluation of systemic, as well as ocular, diseases. Although NLR, PLR, and SII provide excellent predictive value for the risk of other chronic diseases, their relationship with age-related cataract is currently unknown. Therefore, the aim of our study was to investigate the relationship between NLR, PLR, SII calculated from blood indices and cataract using nationally representative NHANES data.

2. Did the cataract group consist of those who had undergone cataract surgery, or were both those who had undergone cataract surgery and those who had not? If the participants had already undergone cataract surgery, what was the purpose of assessing NLR, SII, and PLR in these individuals, and what would this do? 

We have carefully considered your question during the design of the project, but for the following reasons, we have used the answer to the questionnaire ‘Have you ever had cataract surgery? as the main diagnostic basis for cataracts.

Because the diagnosis of cataract as given in the NHANES database can only be determined by whether or not one has undergone intrinsic surgery, such as in the currently published article:

Association between Healthy Eating Index-2015 and Age-Related Cataract in American Adults: a Cross-Sectional Study of NHANES 2005-2008

Association between Life's Essential 8 and cataract among US adults

Selenium intake help prevent age-related cataract formation: Evidence from NHANES 2001-2008

And as we also highlighted in the article, we can be more confident that self-reporting of cataract surgery is more meaningful for cataract diagnosis due to the trend in recent years showing a decrease in visual thresholds for cataract surgery regardless of the level of preoperative vision. The use of NLR, PLR and SII, which are novel inflammatory biomarkers that reflect both the local immune response and the overall level of inflammation in the body. It is valuable for cataract diagnosis and prevention.

3. How do the authors explain the lower risk of cataract in those with BMI <30? 

Our study, however, did find such results, and this explanation may be related to the relationship between inflammation and lipid markers.

An inverted U-shaped relationship between SII and hyperlipidaemia was found in a NHANES study with an inflection point of 479.15 (PMID: 36904176).

Another study found a negative linear relationship between SII and triglycerides (PMID: 36846196)

This suggests that people with a high BMI may also have higher levels of lipids, and that higher levels of lipids, in turn, show a negative correlation with levels of inflammation in the body, which may plausibly explain the lower risk of cataracts in people with a BMI of less than 30.

4. The authors should provide insight into how NLR and SII can be used in clinical practice for cataract. Should everyone with elevated levels of these parameters be screened for cataract? If so, these parameters will increase in numerous chronic and acute inflammatory conditions and are not specific to cataracts. The authors consider how to overcome this handicap. In addition, older age is more associated with cataract risk than these parameters, and if they are to be used for cataract screening, it seems more cost-effective to screen the elderly population. If they are recommended for diagnosis, it is impossible to diagnose cataract by evaluating these parameters. Because increases in these parameters can be detected in many people due to many reasons. 

We did not do this study with the intention of using NLR and SII as cataract-specific indicators, and it is true that they are not cataract-specific. We were mainly exploring NLR and SII as risk factors for cataract. Because of the high incidence of cataract, the significance of NLR and SII is to provide clinicians with new ideas, mainly in the early prevention and intervention of cataract, which is still monitored by ophthalmological specialised equipment. Another important implication is that it strengthens the association between ophthalmology and internal medicine and promotes the integration of the disciplines.

Regarding your suggestion that age is more associated with cataract risk, this was corrected for in our regression analyses, but the results found the NLR and SII were still independent risk factors for cataract, so it can be confidently assumed that the NLR and SII play a key role in the development of cataract.

5. Ratios such as NLR and SII are calculated from data obtained from complete blood counts and are continuously var

---

## [Decision Letter · Decision Letter 1]

25 Oct 2024

Predictive role of the peripheral blood inflammation indices neutrophil-to-lymphocyte ratio (NLR), platelet-to-lymphocyte ratio (PLR), and systemic immunoinflammatory index (SII) for age-related cataract risk

PONE-D-24-25660R1

Dear Dr. Kang,

We’re pleased to inform you that your manuscript has been judged scientifically suitable for publication and will be formally accepted for publication once it meets all outstanding technical requirements.

Kind regards,

Aleksandra Klisic

Academic Editor

PLOS ONE

Additional Editor Comments (optional):

Reviewers' comments:

Reviewer's Responses to Questions

**Comments to the Author**

1. If the authors have adequately addressed your comments raised in a previous round of review and you feel that this manuscript is now acceptable for publication, you may indicate that here to bypass the “Comments to the Author” section, enter your conflict of interest statement in the “Confidential to Editor” section, and submit your "Accept" recommendation.

Reviewer #2: All comments have been addressed

Reviewer #3: All comments have been addressed

2. Is the manuscript technically sound, and do the data support the conclusions?

Reviewer #2: (No Response)

Reviewer #3: Yes

3. Has the statistical analysis been performed appropriately and rigorously? 

Reviewer #2: (No Response)

Reviewer #3: Yes

4. Have the authors made all data underlying the findings in their manuscript fully available?

Reviewer #2: (No Response)

Reviewer #3: Yes

5. Is the manuscript presented in an intelligible fashion and written in standard English?

Reviewer #2: (No Response)

Reviewer #3: Yes

6. Review Comments to the Author

Reviewer #2: (No Response)

Reviewer #3: I would like to thank the authors for their satisfactory responses. They have addressed all comments appropriately.

7. PLOS authors have the option to publish the peer review history of their article (what does this mean?). If published, this will include your full peer review and any attached files.

Reviewer #2: No

Reviewer #3: No

---

## [Editor Report · Acceptance letter]

7 Nov 2024

PONE-D-24-25660R1 

PLOS ONE

Dear Dr. Kang, 

I'm pleased to inform you that your manuscript has been deemed suitable for publication in PLOS ONE. Congratulations! Your manuscript is now being handed over to our production team.

Kind regards, 

on behalf of

Dr. Aleksandra Klisic 

Academic Editor

PLOS ONE